# Transcriptomic Analyses Reveal Key Genes Involved in Pigment Biosynthesis Related to Leaf Color Change of *Liquidambar formosana* Hance

**DOI:** 10.3390/molecules27175433

**Published:** 2022-08-25

**Authors:** Yanjun Li, Yang Zhou, Hong Chen, Chen Chen, Zemao Liu, Chao Han, Qikui Wu, Fangyuan Yu

**Affiliations:** 1Collaborative innovation Centre of Sustainable Forestry in Southern China, Faculty of Forest Science, Nanjing Forestry University, Nanjing 210037, China; 2Faculty of Agriculture and Biotechnology, Zhejiang University, Hangzhou 310058, China; 3State Forestry and Grassland Administration Key Laboratory of Silviculture in Downstream Areas of the Yellow River, Faculty of Forestry, Shandong Agricultural University, Taian 271018, China

**Keywords:** *Liquidambar formosana*, anthocyanins biosynthesis pathway, RNA-Seq, DEGs, leaf coloration

## Abstract

*Liquidambar formosana* Hance has a highly ornamental value as an important urban greening tree species with bright and beautiful leaf color. To gain insights into the physiological and molecular mechanisms of *L. formosana* leaf color change, the leaves of three different clones were sampled every ten days from October 13, 2019, five times in total, which are S1, S2, S3, S4 and S5. Transcriptome sequencing was performed at S1 and S4. The chlorophyll content of the three clones decreased significantly, while the anthocyanins content of the three clones increased significantly in the coloring stage. The anthocyanins content of clone 2 was far more than that of the other two clones throughout the period of leaf color change. The transcriptome analysis showed that six DEGs related to anthocyanins biosynthesis, including *CHS* (chalcone synthase), *CHI* (chalcone isomerase), *F3′H* (flavonoid 3′-hydroxylase), *DFR* (dihydroflavonol 4-reductase), *ANS* (anthocyanidin synthase) and *FLS* (flavonol synthase), were found in three clones. Clone 2 has another three DEGs related to anthocyanins biosynthesis, including *PAL* (Phenylalanine ammonia-lyase), *F3′5′H* (flavonoid 3′,5′-hydroxylase) and *UFGT* (flavonoid 3-O-glucosyltransferase). We lay a foundation for understanding the molecular regulation mechanism of the formation of leaf color by exploring valuable genes, which is helpful for *L. formosana* breeding.

## 1. Introduction

*Liquidambar formosana* Hance, a deciduous ornamental tree species, is widely distributed in Southeast Asia [1,2]. This plant is one of the main afforestation tree species in southern China, which exhibits strong adaptability, fire resistance and high ecological benefits [3]. *L. formosana* has great potential in landscaping and gardening, owing to the fact that its leaf color changes seasonally from green to yellow or red in October [4]; therefore, studying the leaf color of *L. formosana* is of great importance for breeding individual clones with good ornamental values.

Autumn leaf coloration is one of the most important horticultural characteristics in nature [5]. Various colored-leaf trees have gradually become the primary choice for landscaping due to their colorful leaves improving the level and ornamental value of the landscape [6]. Leaf color is affected by the accumulation of different types and quantities of pigments, such as flavonoids and anthocyanins [7]. Flavonoids are able to regulate pigmentation and photo-protection [8]. Anthocyanins are an important class of flavonoids that represent a large group of plant secondary metabolites. They are glycosylated polyphenolic compounds with a range of colors varying from orange, red and purple to blue in flowers, seeds, fruits and vegetative tissues [9]. Anthocyanins are involved in defense responses against pathogens, protecting plants from strong light and UV radiation [10]. They have antioxidant properties to protect plants against various biotic and abiotic stresses [11]. In addition, anthocyanins accumulate in the flowers and fruits of plants and are responsible for their rich color attracting pollinators and seed dispersers [12]. They were catalyzed by complex enzymes from phenylpropanoid and flavonoid biosynthetic pathways [13]. A wide range of constructive genes was involved in the anthocyanin’s biosynthesis. In *Arabidopsis thaliana*, the structural enzymes in the anthocyanins biosynthetic pathway include the upstream components chalcone synthase (*CHS*), chalcone isomerase (*CHI*) and the downstream components dihydroflavonol 4-reductase (*DFR*), anthocyanidin synthase (*ANS*) and others [14]. The structural genes involved in anthocyanins biosynthesis have been identified in genetic analyses of some plants, such as cucumber and sainfoin flowers [15,16]. To date, little is known about the molecular regulatory mechanism of the key genes underlying leaf color change in *L. formosana*. Many studies on *L. formosana* have explored the relationship between leaf color change and environmental factors by measuring the content changes of chlorophyll and other related pigments. Studies of Hu and Liu have pointed out that temperature is the main factor affecting the accumulation of pigment in *L. formosana* leaves [17,18]. This study can enrich the knowledge about the molecular mechanisms of *L. formosana*.

Understanding of the metabolic pathways involved in leaf color change during *L. formosana* development requires the exploration of functional genes. RNA sequencing, based on deep sequencing, has been widely used for gene discovery and analysis of specific genes [19]. The results of this study not only accessed key genes in the anthocyanins biosynthetic pathway of three clones of *L. formosana* in autumn, but also aimed to explore the molecular mechanism of its color formation, thereby providing a theoretical basis for *L. formosana* molecular breeding.

## 2. Results

### 2.1. Dynamic Patterns of Chlorophyll, Carotenoid and Anthocyanin Content

The total chlorophyll, carotenoid and anthocyanins in leaves varied significantly at different coloration stages (Figure 1). With increased redness in leaf color, the total chlorophyll levels decreased (Figure 1a), while anthocyanin levels increased (Figure 1c) and carotenoid levels fluctuated (Figure 1b). For the total chlorophyll, the contents of the three clones at the S1 stage were 2.64 mg/g, 2.87 mg/g and 2.96 mg/g, respectively (Figure 1a), while the contents at the S4 stage were 1.78 mg/g, 1.82 mg/g and 0.52 mg/g, respectively. When compared with S1, the total chlorophyll decreased by 32.6%, 36.6% and 82.4%, respectively. For carotenoids (Figure 1b), the contents of the three clones at the S1 stage were 0.17 mg/g, 0.14 mg/g and 0.09 mg/g, respectively, while content at the S4 stage were 0.02 mg/g, 0.08 mg/g and 0.05 mg/g, respectively. When compared with S1, carotenoids decreased by 88.2%, 42.8% and 44.4%, respectively. For anthocyanins (Figure 1c), levels of the three clones at the S1 stage were 10.75 U/g, 18.20 U/g and 15.57 U/g, respectively, while levels at S4 stage were 24.34 U/g, 61.59 U/g and 20.14 U/g, respectively. Compared with S1, anthocyanins increased by 2.26, 3.38 and 1.29 times. In addition, the anthocyanins levels of clone 2 were far more than that of the other two clones throughout the period of leaf color change.

### 2.2. De Novo Assembly of L. formosana Transcriptome

To explore changes in transcription, a total of 18 samples of three clones at S1 and S4 stages of *L. formosana* were selected, consisting of three clones with three replicates per clone. A total of 936,385,220 raw reads with a total of 141,394,168,220 bp were obtained. After removing the adaptor and low-quality reads, a total of 927,523,280 clean reads with a total of 136,254,863,719 bp were obtained from the eighteen sequencing libraries for further analysis (Appendix A).

### 2.3. Gene Annotation and Classification

In total, 196,890 unigenes (99.65% of the 197,577 total unigenes) were identified by BLASTx (E-value < 1 × 10^−5^) in at least one of the GO, COG and KEGG databases. In total, there were 82,229 (41.62%), 88,492 (44.79%) and 55,770 (28.23%) unigenes that had annotated to the GO, COG and KEGG databases, respectively.

To analyze the functions of DEGs, we used the GO annotation term to enrich the DEGs between S1 and S4 of the three clones. DEGs were divided into three ontologies: molecular function, cellular component and biological process (Appendix A). In clone 1, for the molecular function, “binding” and “catalytic activity” were the frequent terms and were associated with 7577 and 7381 DEGs, respectively. For the cellular component, the DEGs were mainly enriched for “cell part”, “membrane part” and “organelle” with 6809, 4891 and 3892 DEGs, respectively. For the biological process, the DEGs were mainly enriched for “cellular process” (6526 DEGs) and “metabolic process” (6055 DEGs). Similarly, in clone 2, for the molecular function, the DEGs were associated with the “binding” (6758 DEGs) and “catalytic activity” (6709 DEGs); for the cellular component ontology, the DEGs were also enriched for genes involved in “cell part” (5692 DEGs), “membrane part” (4401 DEGs) and “organelle” (3191 DEGs); for the biological process ontology, the DEGs were mainly enriched for “cellular process” (5586 DEGs) and “metabolic process” (5173 DEGs). In clone 3, for the molecular function, the DEGs were associated with the “binding” (6621 DEGs) and “catalytic activity” (6090 DEGs); for the cellular component ontology, the DEGs were also enriched for genes involved in “cell part” (4718 DEGs), “membrane part” (4335 DEGs) and “organelle” (2621 DEGs); for the biological process ontology, the DEGs were mainly enriched for “cellular process” (4852 DEGs) and “metabolic process” (4357 DEGs).

A total of 88,492 unigenes were assigned to 23 COG classifications, with the majority (44,335, 50.10%) in “Function unknown”, followed by “Translation, ribosomal structure and biogenesis” (7038, 7.95%) and “Posttranslational modification, protein turnover, chaperones” (6378, 7.21%) (Appendix A).

A total of 55,770 unigenes were assigned to six KEGG categories and 20 sub-categories (Figure 2). “Metabolism” accounted for the highest proportion, most of which were involved in “Carbohydrate metabolism (6019, 10.79%) and “Amino acid metabolism” (3729, 6.69%). In “Genetic information processing”, “Translation” had the highest number of unigenes (8801, 15.78%), followed by “Folding, sorting and degradation” (3939, 7.06%). In addition, “Transport and catabolism” had a high proportion (3360, 6.02%).

### 2.4. Analysis of DEGs

To analyze the dynamic expression patterns of the specific genes in leaf color change, the differences in transcriptome profiles between the S1 and S4 stages of different clones were compared. The Transcripts Per Million (TPM) values were statistically analyzed to select different unigenes by using the DESeq method [20] (Appendix A).

We compared the DEGs of the three clones (S1-vs-S4). In clone 1, we identified 20,020 DEGs with 16,005 upregulated and 4015 downregulated (Figure 3a,b). In clone 2, there were 17,000 DEGs with 12,941 upregulated and 4059 downregulated (Figure 3a,c). Similarly, there are 17,397 DEGs in clone 3, of which 11,418 DEGs were upregulated and 5979 DEGs were downregulated (Figure 3a,d). Based on these analyses, the DEGs were found and significantly expressed in all three clones, while some of DEGs in clone 2 were the most significant (Figure 3c).

### 2.5. KEGG Pathway Enrichment Analysis of DEGs

In this study, we carried out an enrichment analysis based on the KEGG database in order to explore the biological functions of these DEGs. A total of 8988 unigenes were assigned to 134 KEGG pathways in clone 1 between S1 and S4 stages. Similarly, a total of 7647 unigenes were assigned to 122 KEGG pathways in the clone 2 (S1-VS-S4) and a total of 6341 unigenes were assigned to 135 KEGG pathways in the clone 3 (S1-VS-S4).

Most of the pathways annotated by the three clones were related to “Metabolism”. In the pathways related to leaf color change, the three clones were significantly enriched in “Phenylpropanoid biosynthesis”, “Flavonoids biosynthesis”, “Isoflavonoid biosynthesis”, “Anthocyanin biosynthesis”, “Phenylalanine metabolism” and other pathways (Figure 4).

### 2.6. Identification of DEGs Related to Anthocyanin Metabolism

Based on annotations in public databases, a total of nine anthocyanin-related genes with significant differential expression were obtained, among which six genes were significantly upregulated in three clones (Table 1). *TRINITY_DN11132_c0_g1*, *TRINITY_DN4277_c0_g1*, *TRINITY_DN18660_c0_g2*, *TRINITY_DN11660_c0_g1*, *TRINITY_DN29005_c0_g1*; *TRINITY_DN17255_c0_g3* were annotated as *CHS*, *CHI*, *F3′H*, *DFR*, *ANS* and *FLS*, respectively. Another three genes, including *PAL*(*TRINITY_DN17802_c0_g4*), *F3′5′H*(*TRINITY_DN28662_c0_g1*) and *UFGT*(*TRINITY_DN3115_c0_g1*), were significantly upregulated in clone 2 (Table 2), indicating that there were more DEGs related to anthocyanin biosynthesis in clone 2.

To further validate the reliability of the RNA-seq results, eight DEGs related to anthocyanin biosynthesis were selected (Figure 5). The relative expression of these key genes was very similar to the RNA-seq results, suggesting that the RNA-seq data and DEG analysis are reliable.

## 3. Discussion

### 3.1. Physiological Mechanism of Leaf Color Change of L. formosana

Seasonal changes in the content of chlorophyll, carotenoids and anthocyanins in higher plants are the main reasons for the changes in leaf color [21]. The main components of chlorophyll are Chl A and Chl B. Chl B is unstable and easily decomposed at low temperatures [18]. Anthocyanins are a general term for a large class of compounds, which are flavonoids [13]. Plant leaf color is closely related to anthocyanins [22]. Under acidic soil conditions, anthocyanins appear red, and under alkaline soil conditions, anthocyanins appear blue [18]. In our study, the total chlorophyll content of all three clones significantly decreased, but the anthocyanin content increased from S1 stage to S4 stage. A similar phenomenon was found in the study by Wen and Chu [5]. In their study, the content of chlorophyll decreased, and the content of anthocyanin increased during the color change of *L. formosana*. Nie et al. reached the same conclusion in the change of leaves of *Cotinus coggygria* in autumn [23] Tao et al. also showed that the change of leaf color of poplar had the same findings [24]. In addition, it was found that clone 2 had the brightest color and the highest anthocyanin content (Figure 1c), which was consistent with the most increased anthocyanin in the “red type” in Wen and Chu’s study [5]. The increase in anthocyanins was the primary reason for the change in leaf color in *L. formosana*, causing the brighter color leaves after the coloration period [25].

However, our study on carotenoids was slightly different from that of Hu et al. [17], who reported that the carotenoid content of *L. formosana* leaves decreased significantly at first but did not change significantly thereafter. The dynamic pattern of carotenoid content during leaf color change of *L. formosana* needs further study.

### 3.2. Genes Involved in the Anthocyanin Biosynthesis Pathway

Anthocyanins are an important class of flavonoids that are widely present in plants [13]. Anthocyanin synthesis is catalyzed by a series of enzymes in the phenylpropanoid and flavonoid pathways [26] (Figure 6). The biosynthesis of anthocyanins is controlled by structural genes and regulatory genes, of which structural genes encode biosynthetic enzymes and play a catalytic role in anthocyanin synthesis [27].

Phenylalanine ammonia-lyase (*PAL*) is an enzyme that catalyzes the first step of the phenylpropanoid metabolic pathway [28]. As early as 1960, Neish confirmed that *PAL* catalyzed the synthesis of anthocyanin [29]. *PAL* catalyzes the conversion of phenylalanine to cinnamic acid, while *C4H* catalyzes the conversion of cinnamic acid to 4-coumaric acid. The conversion of 4-coumaric acid to 4-coumaroyl-CoA is catalyzed by *4CL* [26]. 4-Coumaroyl-CoA can generate anthocyanins through flavonoid metabolism, which are important components of flower, fruit and leaf color in plants. Moreover, the synthesis of these substances is closely related to *PAL* activity [30,31,32]. In our study, one *PAL* (*TRINITY_DN17802_c0_g4*) gene in clone 2 was significantly upregulated at S4 stage, and the expression pattern was confirmed in the qRT-PCR analyses; however, we did not find any DEGs annotated as *C4H* and *4CL*, which encode the enzymes required for the production of 4-coumaroyl-CoA.

The first committed enzyme in the flavonoid pathway is Chalcone synthase (*CHS*), a polyketide synthase, mediating the synthesis of naringenin chalcone from 4-coumaroyl-CoA and malonyl-CoA [13]. Then, naringenin chalcone is isomerized by chalcone isomerase (*CHI*) to naringenin, the direct precursor of all flavonoid substances [8]. *CHS* is an important regulatory gene located upstream in the flavonoid biosynthesis pathway, and its overexpression may positively affect the expression of downstream chalcone isomerase (*CHI*) genes that affect the production of flavonoids [33,34]. In our study, both *CHS* (*TRINITY_DN11132_c0_g1*) and *CHI* (*TRINITY_DN4277_c0_g1*) gene showed significantly higher expression levels in three clones at the S4 stage. This result indicated that *CHS* does positively regulate *CHI* expression during the leaf color change of *L. formosana*. At the same time, we also found a significantly upregulated flavonol synthase (*FLS*) gene (*TRINITY_DN17255_c0_g3*) in three clones, which encodes the enzyme that catalyzes naringenin chalcone to flavonols [30]. Flavanone 3-hydroxylase (*F3H*), which belongs to the OGD family, converts naringenin into dihydrokaempferol that can be further hydroxylated by flavonoid 3′-hydroxylase (*F3**′H*) or flavonoid 3′,5′-hydroxylase (*F3**′5**′H*) into two other dihydroflavonols, dihydroquercitin or dihydromyricetin, respectively [35,36]. *F3**′H* and *F3**′5**′H* are the key enzymes determining the structures of anthocyanins, and therefore, they affect color formation [37]. In this study, none of the DEGs were annotated as *F3H*; however, the DEG *TRINITY_DN18660_c0_g2*, which was annotated as *F3′H*, showed a significantly higher expression level at the S4 stage than at the S1 stage in three clones. Another upregulated *F3′5′H* gene (*TRINITY_DN28662_c0_g1*) was screened from clone 2. This result was similar to a study that determined the pathway by which red longan (*Dimocarpus longan*) fruits were produced. Yi et al. revealed that genes related to enzymes leading up to dihydromyricetin were significantly upregulated in red pericarp longan fruits [38]. This may be the reason for the change in leaf color in *L. formosana* and may also be responsible for the bright red color of clone 2 in autumn.

Next, the three dihydroflavonols are reduced to colorless leucoanthocyanidins by dihydroflavonol 4-reductase (*DFR*). Anthocyanidin synthase (*ANS*), which belongs to the OGD family, catalyzes the synthesis of corresponding colored anthocyanidins [39]. Nakatsuka et al. showed that *ANS* gene mutations could cause gentian flowers to turn white [40]. In this study, two upregulated genes, *DFR* (*TRINITY_DN11660_c0_g1*) and *ANS* (*TRINITY_DN29005_c0_g1*) genes were excavated, and their expression levels increased significantly in all three clones at the S4 stage. In the end, anthocyanidins are decorated and glycosylated by various members of the glycosyltransferase enzyme family, for instance, flavonoid 3-O-glucosyltransferase (*UFGT*) [9]. We found that the transcript level of *UFGT* (*TRINITY_DN3115_c0_g1*) in clone 2 was higher at the S4 stage than at the S1 stage. This result indicated that the biosynthesis of anthocyanin compounds is maintained at high levels in clone 2 at the S4 stage. The higher expression levels of *PAL*, *CHS*, *CHI*, *F3′H*, *F3′5′H*, *DFR*, *ANS*, *UFGT* and *FLS* in red leaf than in green leaf of *L. formosana* suggested that these genes are responsible for leaf color formation.

## 4. Materials and Methods

### 4.1. Plant Materials and Growth Conditions

The clones were selected; the trees originated from Qimen county, Anhui province and were vegetatively propagated by grafting. They were planted in the Practice Forest Farm of Nanjing Forestry University (located in XiaShu town, JuRong County, Jiangsu province 32°07′ N, 119°13′ E). In mid-October, 2019, three clones with the most representative coloration effects were selected for sampling. Beginning October 13th, fresh leaves were collected approximately every ten days. The leaves were immediately frozen in liquid nitrogen and stored at −80 °C until use. The five developmental stages were defined according to the time of collection: S1, the green leaf stage; S2, leaf with red margin stage; S3, leaf with red range-expanding stage; S4, the red leaf stage; S5, the red faded stage (Figure 7).

### 4.2. Analysis of Pigment Content

Chlorophyll and carotenoid content of leaves from three clones were measured on the basis of the procedure described by Lichtenthaler and Wellburn [41]. Approximately 0.2 g of fresh samples were ground until no visible tissue in 2 mL of 95% ethanol with a small amount of quartz sand. Then add another 10 mL of 95% ethanol, grind into a homogenate, filter and dilute to 25 mL with ethanol. Finally, the chlorophyll extract was measured by spectrophotometer at the absorption wavelengths of 665 nm, 649 nm and 470 nm. The measurements were performed with three biological replicates.
(1)$$\mathrm{Chlorophyll}\;A\;\mathrm{concentration}\;(\mathrm{mg}/L)=13.95OD_{665}-6.88OD_{649}$$
(2)$$\mathrm{Chlorophyll}\;B\;\mathrm{concentration}\;(\mathrm{mg}/L)=24.96OD_{649}-7.32OD_{665}$$
Total chlorophyll concentration (mg/L) = Chl A + Chl B(3)
(4)$$\mathrm{Carotenoid}\;\mathrm{concentration}\;\mathrm{Car}\;(\mathrm{mg}/L)=(1000OD_{470}-2.05\;\mathrm{Chl}\;A-114.8\;\mathrm{Chl}\;B)/245$$
Pigment content (mg/g) = (pigment concentration ∗ extraction liquid volume ∗ dilution ratio)/sample fresh weight(5)

The content of anthocyanins was determined by the hydrochloric acid-ethanol extraction method [15,42]. In total, 3 g samples were taken and divided into three parts. Anthocyanins were extracted in 10 mL ethanol (containing 1% hydrochloric acid) for 4 h at 32 °C in darkness. The samples were centrifuged at 5000 RPM for 10 min. Supernatants were taken to measure the absorbance at 520 nm, which is the absorbance OD value, repeated three times. Taking 0.1 OD value of fresh weight per gram of leaves in 10 mL extract as a pigment range U, the relative content of anthocyanin is as follows:(6)$$\mathrm{Anthocyanin}\;\mathrm{content}\;(U/g)=OD_{520}/0.1$$

### 4.3. RNA Extraction, cDNA Library Construction and Sequencing

The *L. formosana* leaves at two stages (S1 and S4) were selected as materials. The leaves of S1 are the green ones before coloration, and the leaves of S4 are the red ones after coloration. There were three clones per stage, and each clone had three biological replicates, resulting in a total of 18 samples. Total RNA was extracted using the Plant RNA Kit (Omega Bio-Tek, Doraville, GA, USA) according to the manufacturer’s instructions. The quantity and quality of total RNA were assessed using a 1% agarose gel and a Nanodrop ND 2000 spectrophotometer (Nanodrop Technologies, Wilmington, DE, USA). Total RNA integrity and concentration were assessed using the Bioanalyzer 2100 RNA 6000 Nano Kit (Agilent Technologies, Santa Clara, CA, USA).

Poly(A) mRNA was isolated from total RNA using the Oligotex mRNA Mini Kit (Qiagen, Inc., Valencia, CA, USA) according to the manufacturer’s instructions. The cDNA library was built using methods previously described by Niu et al. [43]. The 18 cDNA libraries were sequenced on the Illumina Hiseq 4000 Sequencing platform (Illumina, Inc., San Diego, CA, USA). The raw data were processed to remove low-quality sequences (more than 50% of reads with Q < 19 bases), adapter-contaminated sequences, and sequences with more than 5% ambiguous base sequences. Clean reads were assembled into unigenes by Trinity software (Trinity Release v2.4.0, MIT and Harvard, Cambridge, MA, USA) [44].

### 4.4. Unigene Annotation and DEG Analysis

Assembled unigenes were aligned to publicly available protein databases, including GO(GeneOntology, http://www.geneontology.org, accessed on 3 April 2022), COG(Clusters of Orthologous Groups of proteins, http://www.ncbi.nlm.nih.gov/COG/, accessed on 3 April 2022) and KEGG(Kyoto Encyclopedia of Genes and Genomes, http://www.genome.jp/kegg/, accessed on 3 April 2022).

Unigenes expression was normalized to Transcripts Per Million (TPM) and the DEGs between different stages were identified with padj < 0.05 and |log2 (foldchange value)| ≥ 1 [45] Next, GO and KEGG enrichment analysis was performed on all DEGs, and a hypergeometric test with a threshold of *p* ≤ 0.05 determined significant enrichment of GO terms and KEGG pathways.

### 4.5. qRT-PCR Validation

Eight key genes involved in anthocyanin biosynthesis were selected for validation by quantitative real-time PCR (QRT-PCR). The primers were designed by Primer Premier 5.0 (Premier Biosoft International, Palo Alto, CA, USA) and the reference gene was 18S ribosomal RNA [46]. All experiments were performed using the StepOne Real-Time PCR System (Applied Biosystems, Foster City, CA, USA) using SYBR Green Dye (Takara, Dalian, China). DEGs were analyzed using the $2^{-\Delta \Delta \mathrm{Ct}}$ method [47]. The experiment was conducted with three biological replicates, and each biological replicates had three technical replicates. The gene-specific primers designed for nine candidate DEGs are listed in Appendix A.

### 4.6. Statistical Analysis

The data analysis included a basic descriptive analysis followed by an analysis of variance (ANOVA). Significant differences were based on Duncan’s test, which were performed using SPSS 23.0 for Windows (SPSS Science, Chicago, IL, USA). The *p*-values less than 0.05 were considered to indicate significance between groups. For the elaboration of graphs, Excel 2019 (Microsoft, Redmond, WA, USA) was used.

## 5. Conclusions

Overall, the regulation mechanism of leaf color in *L. formosana* was firstly carried out by physiology and RNA-seq. It was found that with increased redness in leaf color, the total chlorophyll levels decreased, while anthocyanin levels increased. The anthocyanins content of clone 2 was far more than that of the other two clones throughout the color-changing period. Six genes, including *CHS*, *CHI*, *F3′H*, *DFR*, *ANS* and *FLS*, play an important role in the anthocyanin’s biosynthesis pathway in three clones. Another three genes, including *PAL*, *F3′5′H* and *UFGT*, were only significantly expressed in clone 2, indicating that there were more DEGs related to anthocyanin biosynthesis in clone 2. Our study will provide molecular information for the selection and breeding of new species of colored-leaf species and provide a reference for the future study of leaf color polymorphisms in *L. formosana*.

## Figures and Tables

**Figure 1 molecules-27-05433-f001:**
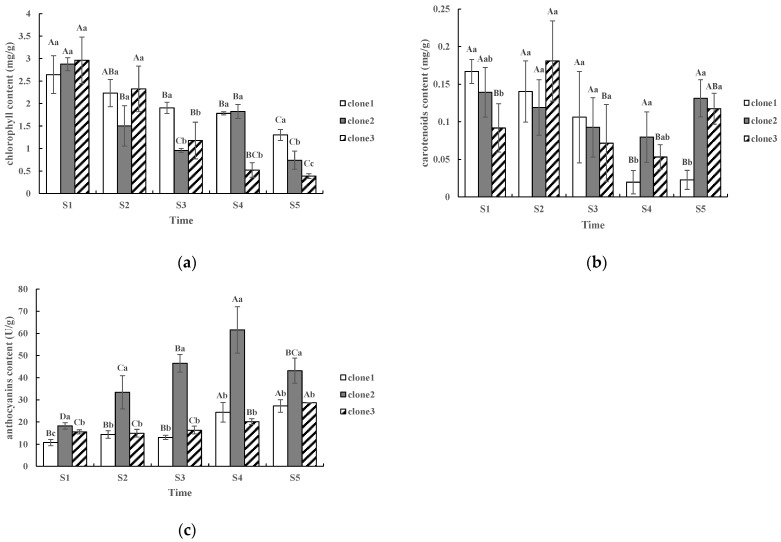
Changes in total chlorophyll, carotenoids and anthocyanin contents during the leaf coloration: (**a**) changes in total chlorophyll content in different clones; (**b**) changes in carotenoids content in different clones; (**c**) changes in anthocyanin content in different clones. Different lowercase letters within each graph indicate significant differences (*p* < 0.05) among different clones, while different capital letters within each graph indicate significant differences among different time periods (*p* < 0.05). There were three biological replicates in each clone at each stage.

**Figure 2 molecules-27-05433-f002:**
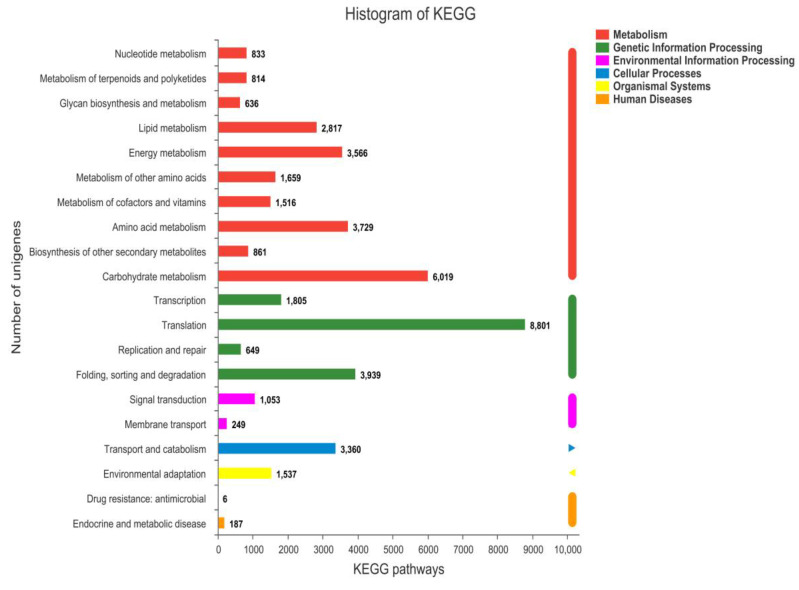
KEGG pathway classification statistics.

**Figure 3 molecules-27-05433-f003:**
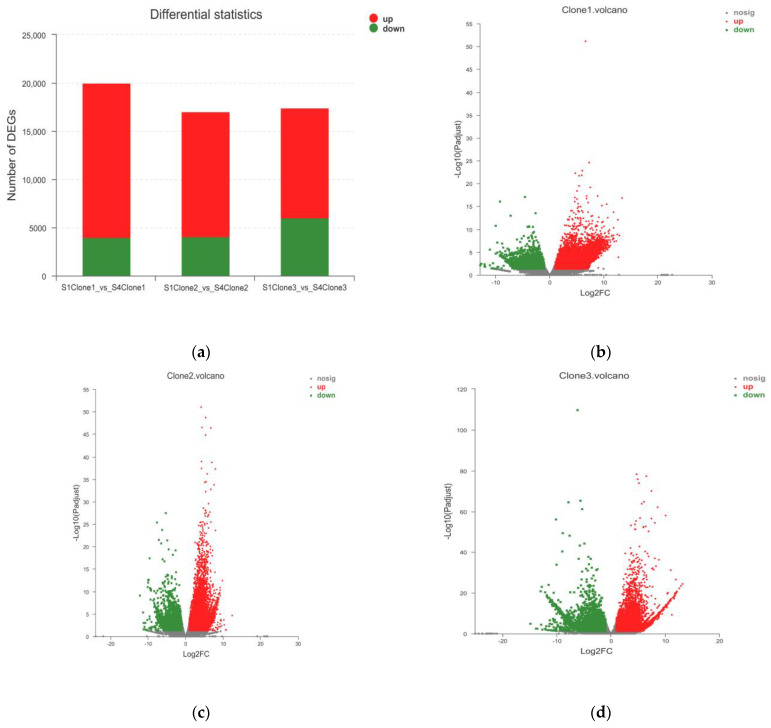
(**a**) Differences in expression levels of different clones (S1-vs-S4); (**b**) differences in expression levels of clone 1 (S1-vs-S4); (**c**) differences in expression levels of clone 2 (S1-vs-S4); (**d**) differences in expression levels of clone 3 (S1-vs-S4). In the volcano map, *y*-axis represents the fold change value of gene expression difference between two stages and *x*-axis shows the statistical test value of gene expression difference; that is, the higher the point is, the more significant the difference is; the farther away from the center, the greater the multiple of difference.

**Figure 4 molecules-27-05433-f004:**
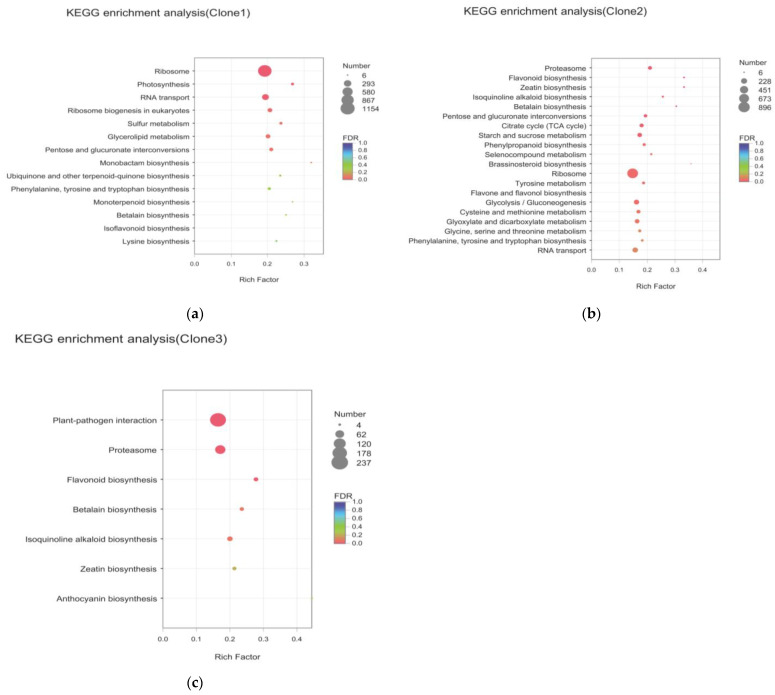
Enrichment of differential genes KEGG in different clones: (**a**) enrichment of differential genes KEGG in clone 1; (**b**) enrichment of differential genes KEGG in clone 2; (**c**) enrichment of differential genes KEGG in clone 3.

**Figure 5 molecules-27-05433-f005:**
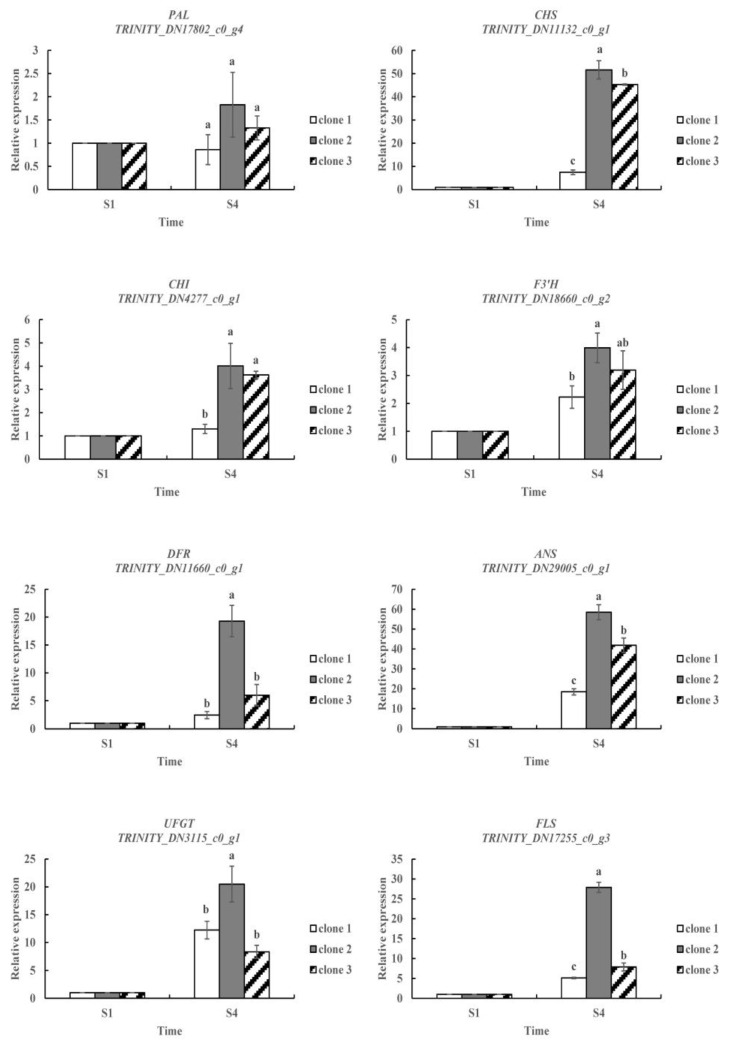
Expression of leaf color-related unigenes of *L. formosana* quantified by qRT-PCR. Bars with different lowercase letters are significantly different (*p* < 0.05). There were three biological replicates in each clone at each stage.

**Figure 6 molecules-27-05433-f006:**
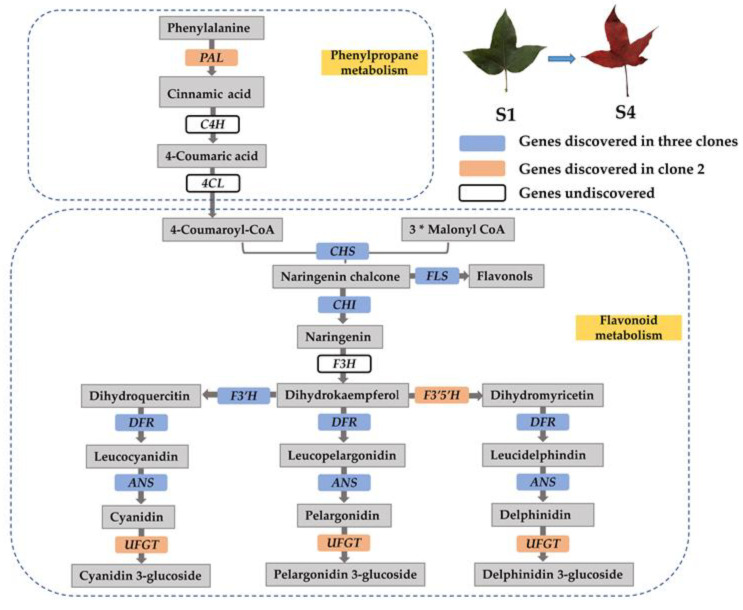
Schematic representation of the anthocyanin biosynthetic pathway. PAL, phenylalanine ammonia lyase; C4H, cinnamate-4-hydroxylase; 4CL, 4-coumaroyl-coA synthase; CHS, chalcone synthase; CHI, chalcone isomerase; F3H, flavanone 3-hydroxylase; F3′H, flavonoid 3′-hydroxylase; F3′5′H, flavonoid 3′,5′-hydroxylase; DFR, dihydroflavonol 4-reductase; ANS, anthocyanidin synthase; UFGT, flavonoid 3-O-glucosyltransferase; FLS, flavonol synthase. The “*” means multiplication.

**Figure 7 molecules-27-05433-f007:**
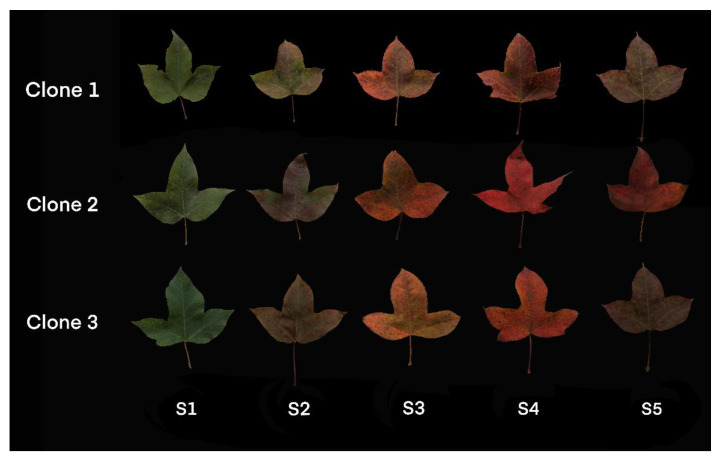
Different leaf colors of *L. formosana* at the five developmental stages.

**Table 1 molecules-27-05433-t001:** DEGs related to anthocyanin biosynthesis in three clones.

Gene ID	Abbreviation	Up/Down	TPM Value					
			S1Clone1	S1Clone2	S1Clone3	S4Clone1	S4Clone2	S4Clone3
*TRINITY_DN11132_c0_g1*	*CHS*	up	5501.80	411.20	420.20	11901.80	3553.40	3352.30
*TRINITY_DN4277_c0_g1*	*CHI*	up	186.20	198.20	210.0	193.2	439.0	468.6
*TRINITY_DN18660_c0_g2*	*F3′H*	up	286.40	3634.70	4101.80	944.1	16367.90	14578.70
*TRINITY_DN11660_c0_g1*	*DFR*	up	3245.80	3301.50	2578.90	802.4	1000.0	1092.40
*TRINITY_DN29005_c0_g1*	*ANS*	up	27.2	19.8	22.4	487.9	868.6	782.4
*TRINITY_DN17255_c0_g3*	*FLS*	up	13.1	23.4	45.0	26.9	128.2	93.9

**Table 2 molecules-27-05433-t002:** DEGs related to anthocyanin biosynthesis in clone 2.

Gene ID	Abbreviation	Up/Down	TPM Value	
			S1Clone2	S4Clone2
TRINITY_DN17802_c0_g4	PAL	up	93.3	216.5
TRINITY_DN28662_c0_g1	F3′5′H	up	1.8	94.5
TRINITY_DN3115_c0_g1	UFGT	up	168.0	468.3

## Data Availability

The datasets generated and analyzed in this study are available at PRJNA837352 (https://www.ncbi.nlm.nih.gov/bioproject/?term=prjna837352), accessed on 12 May 2022.

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
