# Peer review of "Transcriptomic Analyses Reveal Key Genes Involved in Pigment Biosynthesis Related to Leaf Color Change of Liquidambar formosana Hance"

_molecules, 2022, doi:10.3390/molecules27175433_

Round 1

Reviewer 1 Report

The article by Yanjun Li et al. is well conceived and the structure is good. However, I wish to make the following suggestions.

Tittle

Title need to be modified. The phrase ‘in relation’ did not fit. Also need to find consensus between pigment synthesis and leaf colour changes.

Introduction

To avoid repeating similar name continuously, the authors should opt to using pronoun where necessary. For examples, in lines 44, 45 and 47, the name anthocyanin was repeatedly and consecutively used. Consider revising.

The word ‘encoded by’ as used in lines 56, 57 and 58 should be expunge and leave only the short name for the enzymes i.e. CHS, CHI, etc.

The sentence on line 61 ‘There are many studies on ……’ is not clear, consider revising.

Line 64. Add ‘metabolic’ before the word ‘pathways’.

Consider merging or revising the sentences on line 67 and 68.

Results

Gene annotation

Line 134 ought to reads as ‘…..in at least one of GO, COG, KEGG…databases.’

Line 156-159, consider revising. Put the numbers/percentages of the COG immediately after the word ‘majority’ as used in the sentences, i.e ’…with the majority (44,335, 50.10%)…’

Discussion

Line 296 pcinnamic acid

Line 335 and 336, consider revising.

Figure 8 is not clear, completely unreadable.

Materials and methods

Plant materials

There is need for details on how you obtained the clones, or cite your previous work that explain the details if available. Was the initial parent plant obtained from seed or vegetatively grown? Add the details in the text for clarity.

RNA extraction

Line 381. The sampling is not clear. Need to expatiate.

qRT-PCR validation

Lines 413 and 414. Need to be rechecked and revised.

Statistical analysis

The text should be revised for better understanding. What the authors mean by Duncan test?

Conclusion

The conclusion needs to be rewrite.

Author Response

Dear reviewer, thank you very much for your general comment. we have carefully read your comments and suggestions on this article. And we have made a reply to each modification (see Word file for details) and made corresponding modifications in the manuscript. Please check.

Reviewer 2 Report

In this paper, the authors use transcriptomic analysis to reveal key genes related to pigment synthesis for leaf color change of Liquidambar formosana Hance. I have some minor comments as follows to improve the quality of this manuscript:

1. In the abstract, please indicate gene name (CHS, CHI, F3’H, DFR, ANS and FLS).

2. Section 2.1: Please describe more about the content of this section. The authors just briefly describe S4 and S5 stages while the following section 2.2 chooses the S1 and S4 stages for transcriptomic analysis.

3. Section 2.2 and Table 1 can move to SI.

4. What is the difference between annotation data from GO, KEGG, COG, etc? In case the authors mainly used annotation from KEGG for further analysis, GO and COG can move to SI.

Author Response

(The authors gave the same response as above.)
